# BinaryFlex: On-the-Fly Kernel Generation in Binary Convolutional Networks

## Abstract

In this work we present *BinaryFlex*, a neural network architecture that learns weighting coefficients of predefined orthogonal binary basis instead of the conventional approach of learning directly the convolutional filters. We have demonstrated the feasibility of our approach for complex computer vision datasets such as ImageNet. Our architecture trained on ImageNet is able to achieve top-5 accuracy of 65.7% while being around 2x smaller than binary networks capable of achieving similar accuracy levels. By using deterministic basis, that can be generated on-the-fly very efficiently, our architecture offers a great deal of flexibility in memory footprint when deploying in constrained microcontroller devices.

## 1 Introduction

Since the success of AlexNet (Krizhevsky et al., 2012), convolutional neural networks (CNN) have become the preferred option for computer vision related tasks. While traditionally the research community has been fixated on goals such as model generalization and accuracy in detriment of model size, recently, several approaches (Iandola et al., 2016; Courbariaux & Bengio, 2016; Rastegari et al., 2016) aim to reduce the model's on-device memory footprint while maintaining high levels of accuracy.

Binary networks and compression techniques have become a popular option to reduced model size. In addition, binary operations offer $\sim$2x speedup (Rastegari et al., 2016) in convolutional layers since they can be replaced with bitwise operations, enabling this networks to run on CPUs. It is also a common strategy to employ smaller convolutional filter sizes (e.g. 5x5, 3x3 or even 1x1) aiming to reduce the model's memory footprint. Regardless of the filter dimensions, they represent a sizeable percentage of the total model's size. We present a flexible architecture that facilitates the deployment of these networks on resource-constrained embedded devices such as ARM Cortex-M processors. Table 1 shows a range of Cortex-M based microcontrollers running at different frequencies and with different on-chip RAM-Flash configurations.

| Model | Core | Frequency | SRAM | Flash |
|---|---|---|---|---|
| STM32F107RCT6TR | Cortex-M3 | 72 MHz | 64 kB | 0.25 MB |
| STM32F101VFT6TR | Cortex-M3 | 36 MHz | 80 kB | 0.768 MB |
| STM32L496VET6 | Cortex-M4 | 80 MHz | 320 kB | 0.5 MB |
| ATSAME53N20A-AU | Cortex-M4 | 120 MHz | 256 kB | 1 MB |
| MKV58F1M0VCQ24 | Cortex-M7 | 240 MHz | 256 kB | 1 MB |
| ATSAMV71N21B-CB | Cortex-M7 | 300 MHz | 384 kB | 2 MB |

**Table 1:** A varitey of commercially available ARM Cortex-M based microcontrollers

We offer the following contributions: (1) a new architecture that generates the convolution filters by using a weighted combination of orthogonal binary basis that can be generated *on-the-fly* very efficiently and offers competitive results on ImageNet. This approach translates into not having to store the filters and therefore reduce the model's memory footprint. (2) The number of parameters needed to be updated during training is greatly reduced since we only need to update the weights and not the entire filter, leading to a faster training stage. (3) We present an scenario where the *on-the-fly* nature of *BinaryFlex* is benefitial when the model does not fit in RAM.

## 2    RELATED WORK

Our work is closely related to the following areas of research.

**CNN for Computer Vision.**    Deep CNNs have been adopted for various computer vision tasks and are widely used for commercial applications. While CNNs have achieved state-of-the-art accuracies, a major drawback is their large memory footprint caused by their high number of parameters. AlexNet (Krizhevsky et al., 2012) for instance, uses 60 million parameters. Denil et al. (2013) have shown significant redundancy in the parameterization of CNNs, suggesting that training can be done by learning only a small number of weights without any drop in accuracy. Such over-parameterization presents issues in testing speed and model storage, therefore a lot of recent research efforts have been devoted to optimizing these architectures so that they are faster and smaller.

**Network Optimization.**    There have been many attempts in reducing model size through use of low-precision model design. Linear quantization is one approach that has been rigorously studied: One direction in quantization involves taking a pre-trained model and normalizing its weights to a certain range. This is done in  Vanhoucke et al. (2011) which uses an 8 bits linear quantization to store activations and weights. Another direction is to train the model with low-precision arithmetic as in Courbariaux et al. (2014) where experiments found that very low precision could be sufficient for training and running models. In addition to compressing weights in neural networks, researchers have also been exploring more light-weight architectures. SqueezeNet (Iandola et al., 2016) uses $1 \times 1$ filters instead of $3 \times 3$, reducing the model to $50\times$ smaller than AlexNet while maintaining the same accuracy level.

**Binary Networks.**    In binary networks, parameters are represented with only one bit, reducing the model size by $32\times$. Expectation BackPropagation (EBP) (Soudry et al., 2014) proposes a variational Bayes method for training deterministic Multilayer Neural Networks, using binary weights and activations. This and a similar approach (Esser et al., 2015) give great speed and energy efficiency improvements. However, the improvement is still limited as binarised parameters were only used for inference. Many other proposed binary networks suffer from the problem of not having enough representational power for complex computer vision tasks, e.g. BNN (Courbariaux & Bengio, 2016), DeepIoT (Yao et al., 2017), eBNN (McDanel et al., 2017) are unable to support the complex ImageNet dataset. In this paper, we also considered the few binary networks that are able to handle the ImageNet dataset, i.e. BinaryConnect (Courbariaux et al., 2015) and Binary-Weight-Networks (BWN) (Rastegari et al., 2016), achieving the later AlexNet level accuracy on ImageNet. BinaryConnect extends the probabilistic concept of EBP, it trains a DNN with binary weights during forward and backward propagation, done by putting a threshold for real-valued weights.

## 3    BINARYFLEX

### 3.1    CONVOLUTION USING BINARY ORTHOGONAL BASIS

Training CNNs for image classification fundamentally consists in learning the set of filters that hierarchically transform an input image to an image label. The filters are usually learn via back-propagation (LeCun et al., 1989). BinaryFlex differs from standard CNNs in the sense that filters are not directly learnt. Instead, they are generated from a weighted linear combination of deterministic orthogonal binary basis. Our architecture learns the per-basis weighting coefficients and uses them to generate the filters at inference time.

We introduce in our architecture a basis generator that outputs orthogonal variable spreading factor (OVSF) codes that are later combined to generate the convolutional filters. This generator has been widely used in W-CDMA based 3G mobile cellular systems[1] to provide multi-user access. Its simplicity, makes it suitable for efficient real-time implementations on power-constrained devices.

The filter generation process using the OVSF basis and the learned weights is illustrated in Figure 1. The OVFS generator outputs a set of binary sequences of length $L$, where $L = 2^l, l \in \mathbb{N}$ that are orthogonal to each other; they get reshaped to match the shape of the desired convolutional filter; and finally linearly combined using the weights obtained during training. The quality of our filter

---

[1]3GPP TS 25.213, v 3.0.0, Spreading and modulation (FDD), Oct. 1999

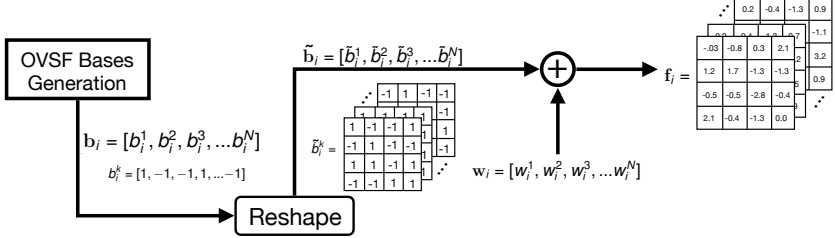

**Figure 1:** From orthogonal binary basis to convolutional filters.

approximation can be measured as:

$$E_k = |f'_k - f_k| = |\sum_{i=0}^{\rho} w_k^i b_k^i - f_k| < \epsilon \qquad (1)$$

Where $\rho$ is the total number of basis to use in order to approximate filter $f_k$, $b_j^i$ is a base and $w_j^i \in \mathbb{R}$ its associated weight. $\epsilon$ is the difference between the the approximated, $f'_k$ and the real filter, $f_k$. Intuitively, $\epsilon \to 0$ as we increase the number of binary basis, $\rho$. In the following sections we will describe how these basis are generated and why it is beneficial for our flexible model with on-the-fly filter generation.

## 3.2 GENERATION OF BINARY ORTHOGONAL BASIS

The design space to generate high-dimensional binary and orthogonal arrays is very broad. Our target platforms constrains this space to those codes that can be generated very eficiently while been applicable to complex image classification tasks. We find OVSF codes to be an optimal solution given those constrains.

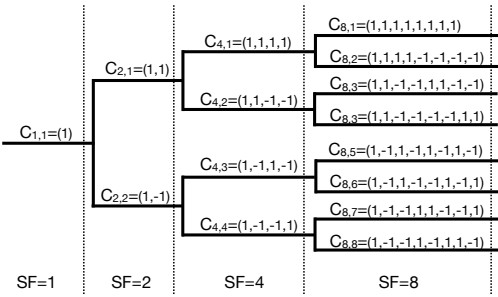

**Figure 2:** Code Tree for OVSF Code Generation

Figure 2 shows the procedure for generating OVSF codes of different lengths as a recursive process in a binary tree (Adachi et al., 1997). Assuming code $C$ with length N on a branch, the two codes on branches leading out of $C$ are derived by arranging $C$ in $(C, C)$ and $(C, -C)$. It is clear from this arrangement that $C$ can only be defined for $N$s that are powers of 2.

Efficient hardware implementations of OVSF code generators have been crucial for designing power-efficient cellular transceivers and given the maturity of W-CDMA standard is a well-studied problem in the wireless community (Andreev et al., 2003; Kim et al., 2009; Rintakoski et al., 2004; Purohit et al., 2013). Availability of such designs makes OVSF codes a suitable choice for efficient on-device filter generation where memory for storing the model comes at a premium.

## 3.3 TRADING COMPUTE FOR MEMORY: ON-THE-FLY BASIS GENERATION

In scenarios with very memory-limited platforms, a network making use of *on-the-fly* basis generation would only be required to store the basis coefficients (the weights) and generate the filters

*on-the-fly*. This flexibility is possible since the generation of binary basis is deterministic, repeatable and can be achieved using simple hardware logic on silicon. In addition, this process can be done with high granularity. For example, the application can maintain just a subset of basis in memory or, given that OVSF codes are conventionally generated recursively from the code tree described in 3.2, the model can store only parts of the basis and generate the remaining parts when required.

When memory footprint is not a concern the application can cache kernel coefficients and make BinaryFlex behave exactly similar to conventional CNNs while a middle-ground approach would be to only store the basis in memory and combine them to recover the kernels. The decision of which approach to take can either be made at design time based on the specification and requirements of the target device but a smart implementation can mix-and-match these approaches at run-time depending on the amount of available resources at inference time.

The deterministic nature of basis generation also gives hardware implementations flexibility in terms of the type of memory that can be used. For on-device learning on very resource-stringent devices this approach allows the implementation to move a big portion of network parameters into ROM instead of RAM resulting in much smaller silicon area and power consumption.

### 3.4 BINARYFLEX ARCHITECTURE OVERVIEW

Our BinaryFlex architecture, Figure 3, is inspired by SqueezeNet(Iandola et al., 2016) and ResNet(He et al., 2015). Macroarchitecturally, it resembles SqueezeNet in the sense that after the initial convolution and max-pooling layers, the rest of the pipeline is comprised of 3 cascades of modules or blocks separated by max-pooling layers. The final elements are a convolutional and pooling layer followed by a softmax stage. Microarchitecture wise, Binaryflex resembles ResNet in the sense the flow of data in the building blocks follow the same pattern as in ResNet's bottleneck architecture. In BinaryFlex we name this building block *FlexModule*.

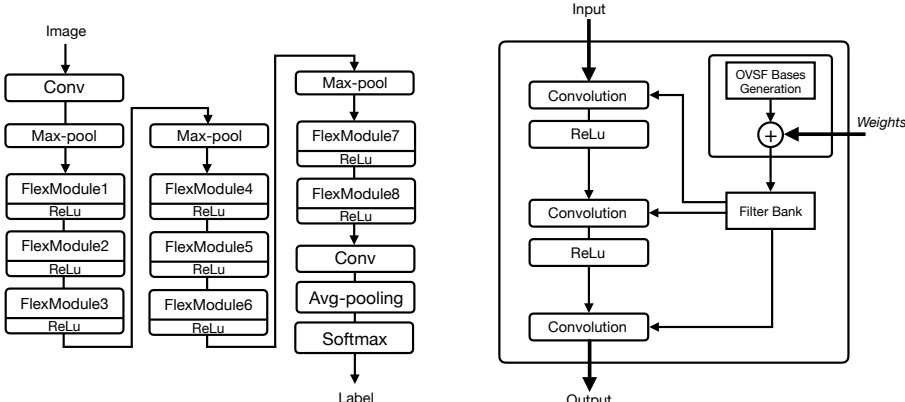

**Figure 3:** BinaryFlex architecture.          **Figure 4:** FlexModule overview.

BinaryFlex is comprised of eight *FlexModules* and no fully connected layer aiming to reduce the number of parameters in our model. We adopted SqueezeNet's strategy of late down-sampling that in summary consists in using stride $s = 1$ in each of the convolution layers in the *FlexModules* and perform downsampling after each cascade of those modules (i.e. after blocks *FlexModule3*, *FlexModule6* and *FlexModule8*). This microarchitecture enables the convolutional layers to have large activation maps. Delaying the downsampling in CNNs leads to better accuracy in certain architectures (He & Sun, 2014).

In isolation, a *FlexModule* looks like Figure 4. Each convolution layer uses a different set of filters from the *filter bank* that has been generated as illustrated in Figure 1. In our BinaryFlex implementation, a per-*FlexModule* OVSF basis-generator is not needed since the basis are deterministic and we only have to decide how many basis to combine in order to create each filter. Further explanation on how the basis are generated and how they are efficiently used during inference can be found in 3.2 and 3.3, respectively.

## 4 EVALUATION

In this section, we evaluate BinaryFlex's performance when compared to binary networks on image classification tasks when deploying in compute and memory constrained platforms. We first present results on different vision problems to demonstrate the generalizability of BinaryFlex. Then we present a detailed study of BinaryFlex's performance on ImageNet, followed by a case study of BinaryFlex's flexibility in memory footprint. The main findings are:

- Binaryflex offers the best *accuracy to size* ratio (ASR) previously seen. One configuration is able to support ImageNet at a reasonable accuracy of 56.5% while being just 1.6 MB.

- BinaryFlex is comparable in accuracy under common image datasets. In particular for ImageNet, BinaryFlex accuracies dominate the spectrum small model sizes (1 MB to 4 MB) only lagging behind in bigger models, resulting in less than 5% accuracy drop.

- Our work demonstrates the feasibility of using a combination of deterministic binary basis as convolutional filters, achieving fairly high accuracy models in image classification tasks.

- BinaryFlex can be easily adjusted to meet the computation and memory requirements very constrained platform (see Table 1). The breadth of the inference operations vs memory size space spanned by our architecture is much wider than that offered by exiting models.

| Layer Type | Output Size | Filter Size/stride | Conv1 Filer | Conv2 Filer | Conv3 Filter | OVSF ratios |
|---|---|---|---|---|---|---|
| input image | 256×256×3 | | | | | |
| Conv (in) | 128×128×64 | 1×3×3×64/1 | | | | |
| MaxPool 1 | 64×64×64 | 1×3×3×1/2 | | | | |
| FlexModule1 | 64×64×128 | | 64×1×1×32 | 32×4×4×64 | 64×1×1×128 | [1.0, 0.25, 1.0] |
| FlexModule2 | 64×64×128 | | 128×1×1×32 | 32×4×4×64 | 64×1×1×128 | [0.5, 0.25, 1.0] |
| FlexModule3 | 64×641128 | | 128×1×1×32 | 32×4×4×64 | 64×1×1×128 | [0.5, 0.25, 1.0] |
| MaxPool 2 | 32×32×128 | 1×3×3×1/2 | | | | |
| FlexModule4 | 32×32×256 | | 128×1×1×64 | 64×4×4×128 | 128×1×1×256 | [0.5, 0.125, 0.5] |
| FlexModule5 | 32×32×256 | | 256×1×1×64 | 64×4×4×128 | 128×1×1×256 | [0.5, 0.125, 0.5] |
| FlexModule6 | 32×32×256 | | 256×1×1×64 | 64×4×4×128 | 128×1×1×256 | [0.5, 0.25, 0.5] |
| MaxPool 3 | 16×16×256 | 1×3×3×1/2 | | | | |
| FlexModule7 | 16×16×512 | | 128×1×1×64 | 64×4×4×128 | 128×1×1×256 | [0.5, 0.125, 0.5] |
| FlexModule8 | 16×16×512 | | 256×1×1×64 | 64×4×4×128 | 128×1×1×256 | [0.5, 0.125, 0.5] |
| Conv (out) | 8×8×1001 | 1×1×1×512/2 | | | | |
| AvgPool | 1×1×1001 | 1×3×3×1/8 | | | | |

**Table 2:** BinaryFlex architectural dimensions and parameters for the 3.4MB model trained on ImageNet. All BinaryFlex configurations for ImageNet maintain the above parameters and only vary the ratios. For MNIST and CIFAR-10 the final layers are slightly modified to accommodate for the number of classes in those datasets.

### 4.1 EXPERIMENTAL SETUP

To train BinaryFlex on ImageNet, we used learning rate 0.1, decade factor 0.1 per 30 epochs and batch size 64. For the case of CIFAR-10 and MNIST we train each BinaryFlex configuration for a total of 90 epochs with the same decay rate as before and batch size of 128. No pre-processing or data augmentation is used. Our architecture is implemented in TensorFlow.

### 4.2 RESULTS

On ImageNet, BinaryFlex is compared to BinaryConnect (Courbariaux et al., 2015) and BNN(Courbariaux & Bengio, 2016); for MNIST and CIFAR-10, BinaryFlex is compared to BinaryConnect and BWN (Rastegari et al., 2016). Several BinaryFlex configurations resulting in different model sizes are trained on ImageNet, CIFAR-10 and MNIST. The fundamental difference between each configuration is the ratios of binary basis used to generate the convolutional filters. Intuitively, combining more basis results in filters that can better capture complex image features. The results in Table 3 show the robustness of BinaryFlex even when using a very small portion of binary basis, resulting in smaller models. When comparing the performance of several BinarFlex configurations against other models for the same dataset, we use *BinaryFlex* appended by the model size in MB, e.g. BinaryFlex-3.6, to refer to a particular BinaryFlex configuration. BinaryFlex-10.3 uses almost all the OVSF to generate the filters, where as BinaryFlex-3.4 uses around 30% and BinaryFlex-0.7 close to 5%. As shown in Table 2, ratios values differ between FlexModules. Our

findings suggest that lower OVSF ratios in deeper layers provide higher model size reductions at the cost of little to none accuracy loss when shallower layer maintain high OVSF ratios.

**ImageNet.** In Table 5, we see BinaryFlex's ability to offer good ASR which are otherwise not possible. We proof that, OVSF basis can provide comparable results to those achieved by BWN at a similar model size even when BinaryFlex has not been fine tuned during training. BinaryFlex can be made very small, as BinaryFlex-1.6 gives a $4.5\times$ reduction in model size but only gives up 4% in accuracy when comparing to BinaryConnect. This small model size of 1.6MB is of critical importance for hardware implementations as many processors (e.g. ARM Cortex processor M7) only allows a maximum combined ROM/RAM space of 2MB. By fitting into ROM/RAM of embedded devices without paging to SD card, BinaryFlex offers great computational advantage, as the timescales of data movement from SD card to RAM can be $1250\times$ longer than that from ROM to RAM (Gregg, 2013).

**Other datasets.** Table 4 shows that BinaryFlex gives competitive results in a general setting, despite not having its architectures optimized for each task. On MNIST, BinaryFlex gives the best accuracy amongst baselines. In addition, with the aims to further reduce memory footprint, we applied post-training 8-bit quantisation to the BinaryFlex models shown in Table 3 for MNIST and CIFAR-10 datasets resulting in models $\sim$75% smaller. Some results are shown in Table 4.

| Dataset | 10.3MB | 7.3MB | 5.5MB | 3.4MB | 2.2MB | 1.6MB | 1.2MB | 0.7MB |
|---|---|---|---|---|---|---|---|---|
| MNIST | 99.6 | 99.6 | 99.6 | 99.3 | 99.5 | 99.5 | 99.5 | 99.1 |
| CIFAR-10 | 89.7 | 88.3 | 88.0 | 86.3 | 85.5 | 84.1 | 82.4 | 79.9 |
| ImageNet | 53.3/77.3 | 50.4/74.8 | 46.7/71.2 | 40.5/65.7 | 36.5/61.9 | 31.5/56.5 | $\times$ | $\times$ |

**Table 3:** Accuracy (%) of Binaryflex in three popular image datasets at different model sizes (i.e. different OVSF ratios). For ImageNet, accuracy values are shown as Top-1/Top-5.

| Model | CIFAR-10 | MNIST |
|---|---|---|
| BinaryConnect | **90.1** / 0.73 MB | 98.7 / 0.35 MB |
| BNN | 89.9 / 0.73 MB | 99.0 / 4.5 MB |
| BinaryFlex-10.5q8 | 88.3 / 2.6 MB | **99.42** / 2.6 MB |
| BinaryFlex-2.2q8 | 84.4 / 0.55 MB | 99.37 / 0.55 MB |
| BinaryFlex-1.2q8 | 79.4 / **0.3 MB** | 98.84 / **0.3 MB** |

| Model | Acc. | Size | ASR |
|---|---|---|---|
| BWN | **56.8 / 79.4** | 7.8 | 10.2 |
| BinaryFlex-7.3 | 50.4 / 74.8 | 7.3 | 10.2 |
| BinaryFlex-3.4 | 40.5 / 65.7 | 3.4 | 19.32 |
| BinaryFlex-2.2 | 36.5 / 61.9 | **2.2** | **28.13** |
| BinaryConnect | 35.4 / 61.0 | 7.8 | 7.8 |

**Table 4:** Accuracy of BinaryFlex and baselines on three simple classification datasets, shown in the format: accuracy % / model size (MB). The naming BinaryFlex-Xq8 refers to a BinaryFlex model of X MB (when representing the weights as 32-bit values) whose weights have been 8-bit quantized.

**Table 5:** Accuracies (%), model sizes (MB) and size-to-accuracy ratios of binary networks on ImageNet shown as Top-1/Top-5 values. Ratios are computed using Top-5 accuracy values.

## 4.3 Case Study: Performance Trade-offs of BinaryFlex under ImageNet.

In this section we elaborate on the performance of BinaryFlex model on ImageNet, as our main goal is to improve the performance of binary architecture on large-scale dataset. There two things we want to emphasize, the flexibility of different model sizes and good memory trade-off. First, by adjusting the number of basis used for each filter generation, we can easily find a model that meets the memory constraint of any device which the model is going to be deployed on. Second, it is very important to take into account both the accuracy and model size when training models, since we do not want to compromise the accuracy too much while reducing the model size. And so, we introduce the concept of *accuracy to size ratio (ASR)* to evaluate the trade-off between accuracies and model sizes. As shown in Figure 5, BinaryFlex not only has a wide range of model sizes, from 10MB to 1MB but also maintains high ASR, ranging from 10 to 35(%/MB), when the model size gets reduced.

## 4.4 Case Study: Flexibility of BinaryFlex

In this case study, we further emphasize the benefit attributed to other on-the-fly generation. The overhead of on-the-fly generation is extremely low. To generate basis with dimension $n$, it only requires $2 \times (2^{log_2 n} - 1)$ operations. And with dedicated hardware, the whole generation can be

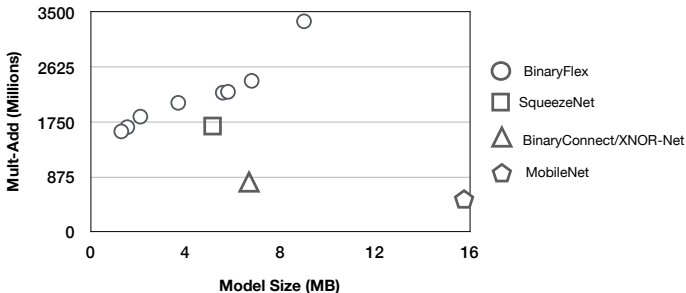

**Figure 5:** Model size and number of operations of BinaryFlex on ImageNet at different configurations. Comparison against other architectures. Circles represent all the trained configurations of BinaryFlex.

done in just one operation cycle. As a result, on-the-fly generation gives rise to substantial memory footprint reduction with just little overhead. In Figure 6, we analyzed the model size and number of Mult-Adds resulting from different number of basis used for filter generation and percentage of basis that are generated during runtime. We examined three different generation configurations: 1) 100% generation – All the binary basis are generated during runtime. 2) 50% generation – Half of the binary basis are generated during during runtime and other half are stored in the memory and retrieved during filter generation. 3) No generation – All the basis are stored in the memory. The on-the-fly generation results in 5X to 10X memory with negligible compute overhead. The gain is even more pronounced for models with large size. This suggests that in scenarios where we have deploy large model on a resource-limited device, we can apply our method to reduce the model size without compromising the accuracy. In other words, we can determine the fraction of basis to be generated during runtime based on the memory and compute resource on the device.

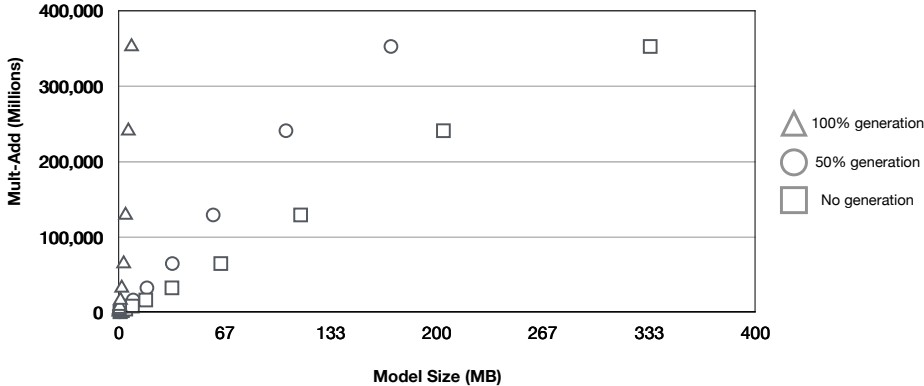

**Figure 6:** Model size and number of operations of BinaryFlex on ImageNet at different configurations when varying the portion of basis generated *on-the-fly*. When off-loading the convolutional filters from the model, i.e. using OVSF basis to generated them on the fly, the model size is drastically reduced.

## 5 DISCUSSION

Here we further describe the main contributions of BinaryFlex and why we believe they are relevant.

Our network architecture is flexible. We could define this as a 2-dimensional property in the sense that it can be split into two distinctive components: (1) reduction of the model size by enabling on-the-fly generation of basis and (2) reduction in the number of forward operations depending on the number of basis we use to generate each filter. The first component could be defined as *on-the-fly* flexibility, $F_{OTF}(T_{RAM}, model_{MB})$, where subscript $_{OTF}$ stands for *on-the-fly*, $T_{RAM}$ is

the available RAM in the target platform and $model_{MB}$ is the size of the BinaryFlex model. This property was first described in 3.3. Figure 7 provides an intuition on how BinaryFlex could be deployed into platforms with limited on-chip memory. The model accuracy is not being affected by $F_{OTF}$.

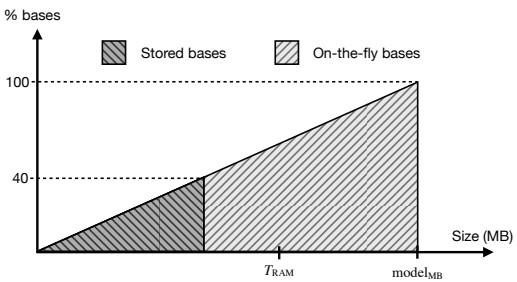

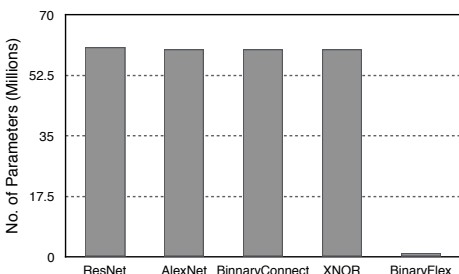

**Figure 7:** Scenario where the model doesn't not fit in the RAM of the target platform.

**Figure 8:** Num. of parameters updated during ImageNet training.

We define the second flexibility component as *operations* flexibility, $F_{OP}(T_{COMP}, A_\epsilon)$, where $T_{COMP}$ is the compute capability of the target platform $T$ and $A_\epsilon$ is the maximum allowed accuracy error allowed in application $A$ (i.e. image classification). Intuitively, the $F_{OP}$ flexibility component enables BinaryFlex to run on compute-constrained devices without having to modify the network architecture, at the cost of reducing the model accuracy. We believe that, unlike other methods that limit their efforts to reduce the complexity of the operations (e.g. by using binary operations), this characteristic of BinaryFlex, $F = \{F_{OTF}, F_{OP}\}$, would lead to a new type of CNNs specifically designed to perform well in a broad range of constrained devices.

In this work we have focused on reducing already small CNNs, in the order of a few MBs, to sub-MB models by avoiding to store the filters of each convolutional layer. This characteristic of BinaryFlex is equally applicable to the other side of spectrum, big models like AlexNet(240MB) or VGG(540MB). This would enable training much larger models while keeping their memory footprint at an order of magnitude less. Figure 6 visually represents this idea.

So far, we have discussed why BinaryFlex's property of *on-the-fly* basis generation leads to a reduction in the model size. In addition to this, our approach dramatically reduces the number of parameters that need to be updated during each training iteration. Figure 8 compares the number of parameters learned in each architecture for the ImageNet classification task. Because BinaryFlex only needs to store the weights for each basis and not the entire filters, the amount of parameters is reduced by an order of magnitude. Implicity to this property is that, since the size of a BinaryFlex model depends on the number of basis used to generate the filters and not on the dimensions of this filters, *FlexModules* in our architecture enables the usage of arbitrarily big filters without translating this into an increase in model memory footprint. We find this novel approach to be the to-go option to enable training with more and bigger filters that better capture relevant image features.

# 6 CONCLUSION

We have introduced BinaryFlex, a small-memory, flexible and accurate binary network which learns through a predefined set of orthogonal binary basis. We have shown that BinaryFlex can be trained on ImageNet, CIFAR-10 and MNIST with good balance of accuracy and model size. On ImageNet, BinaryFlex models are able to reduce the size by $4.5\times$ without great compromise in accuracy. Further, we have introduced techniques such as on-the-fly basis generation, allowing powerful management of inference-time overhead on low-resource devices.

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
