# OpenReview forum: "BinaryFlex: On-the-Fly Kernel Generation in Binary Convolutional Networks"
_ICLR.cc/2018/Conference — Reject_

### Official Review · AnonReviewer1 · 2017-11-27

**Rating:** 5
**Confidence:** 3

**Review:**

The paper presents a binary neural network architecture that operated on predefined orthogonal binary basis. The binary filters that are used as basis are generated using Orthogonal Variable Spreading Factor.
Because the filters are weighted combinations of predefined basis, only the weights need to be trained and saved. The network is tested on ImageNet and able to achieve top-5 accuracy of 65.9%.

The paper is clearly written. Few mistakes and questions:
Is Equation 2 used to measure the quality of the kernel approximation?

In Figure 2, what is Sparse layer? Is it FlexModule?

In 4.1 Results Section, the paper states that “On ImageNet, BinaryFlex is compared to BinaryConnect and BinaryNeuralNet; otherwise, BinaryFlex is compared to BinaryConnect and BNN.” It should be Binary Weight Network instead of BinaryNeuralNet.

Based on Results in Table 1, BinaryFlex is able to reduce the model size and provide better accuracy than BinaryConnect (2015). However, the accuracy results are significantly worse than Binary-Weight-Network (2016). Could you comment on that? The ImageNet results are worrying, while BNN (7.8Mb) achieves 79.4%, this BinaryFlex (3.4Mb) achieves 65.7%. The accuracy difference is huge.

---

> ### Author Response · Authors · 2018-01-05
> **Post-review modifications**
>
> Thank you for your comments and issues raised in our submission. Many of the issues were raised by all the reviewers, we believe to have addressed them all.
>
> We have provided a fairer comparison between BinaryFlex and BWN on ImageNet when both architectures have similar model sizes. The difference in accuracy terms is less than 5%.

---

### Official Review · AnonReviewer3 · 2017-11-27
**BinaryFlex: On-the-Fly Kernel Generation in Binary Convolutional Networks**

**Rating:** 5
**Confidence:** 3

**Review:**

The paper proposes a neural net architecture that uses a predefined orthogonal binary basis to construct the filter weights of the different convolutional layers. Since only the basis weights need to be stored this leads to an exponential reduction in memory. The authors propose to compute the filter weights on the fly in order to tradeoff memory for computation time. Experiments are performed on ImageNet, MNIST, CIFAR datasets with comparisons to BinaryConnect, Binary-weight-networks and studies showing the memory vs time vs accuracy tradeoff.

Positives
- The idea of using a predefined basis to estimate filter weights in a neural network is novel and leads to significant reduction in memory usage.

Negatives
- The proposed method seems significantly worse than other binary techniques on ImageNet, CIFAR and SVHN. On Imagenet in particular binary-weight-network is 21% better at only 2x the model size. Would a binary-weight-network of the same model size be better than the proposed approach? It would help to provide results using the proposed method with the same model size as binary-weight-networks on the different datasets.
- The citation to binary-weight-networks is missing.
- The descriptions in section 3.3, 3.4 need to be more rigorous. For instance, how many basis weights are needed for a filter of size N. Does N need to be a power of 2 or are extra dimentions from the basis just ignored?

---

> ### Author Response · Authors · 2018-01-05
> **Review update**
>
> We appreciate your comments. We have now submitted a new version of our paper and addressed the issues raised during the reviewing period.
>
> The standard OVSF codes are a power of 2 arrays of 1s and -1s. Using other configurations, e.g. leading to 5x5 kernels, is a possibility that we haven't explored at this stage.

---

### Official Review · AnonReviewer2 · 2017-11-28
**This paper proposes using a set of orthogonal basis and their combination to represent convolutional kernels. To learn the set of basis,  the paper uses an existing algorithm (OSVF). The paper lacks details to clearly understand what is being done and how to properly interpret results.**

**Rating:** 3
**Confidence:** 4

**Review:**

This paper proposes using a set of orthogonal basis and their combination to represent convolutional kernels. To learn the set of basis,  the paper uses an existing algorithm (OSVF)

-- Related Work

Related work suggests there is redundancy in the number of parameters (According to Denil et al) but the training can be done by learning a subset directly without drop in accuracy. I am not really sure this is strictly correct as many approaches (including Denil et al) suggest the additional parameters are needed to help the optimization process (therefore hard to learn directly a small model).

As in the clarity point below, please be consistent. Acronyms are not properly defined.


-- Method / Clarity

It is nice to read section 3.1 but at the same time probably redundant as it does not add any value (at least the first two paragraphs including Eq. 1). Reading the text, it is not clear to me why there is a lower number of parameters to be updated. To the best of my understanding so far in the explanation, the number of parameters is potentially the same but represented using a single bit. Rephrasing this section would probably improve readability.

Runtime is potentially reduced but not clear in current hardware.

Section 3.2 is nice as short overview but happens to take more than the actual proposal (so I get lost).

Figures 2 and 3. I am surprissed the FlexModule (a building block of BinaryFlex) is not mentioned in the binaryflex architecture and then, sparse blocks are not defined anywhere. Would be nice to be consistent here. Also note that filter banks among other details are not defined.


Now, w and b in eq 2 are meant to be binary, is that correct? The text defines them as real valued so this is confusing.

- From the explanations in the text, it is not clear to me how the basis and the weights are learned (except using backprop). How do we actually generate the filter bank, is this from scratch? or after some pretraining / preloaded model? What is the difference between BinaryFlex models and how do I generate them when replicating these results? It is correct to assume f_k is a pretrained kernel that is going to be approximated?




-- more on clarity


I would also appreciate rephrasing some parts of the paper. For instance, the paragraph under section 4.1 is confusing. There is no consistency with namings / acronyms and seems not to be in the right order. Note that the paragraph starts talking about ImageNet and then suggests different schedules for different datasets. The naming for state-of-the-art methods is not consistent.
Also note that acronyms are later used (such as BWN) but not defined here. This should be easy to improve.

I guess Figurre 4 needs clarification. What are the axis? Why square and cercles? Same for Figure 5.

Overall, text needs reviewing. There are typos all over the text. I think ImageNet is not a task but classification using ImageNet.

-- Results

I find it hard to follow the results. Section 4.1.1 suggests accuracy is comparable when constraints are relaxed and then only 7% drop in accuracy for a 4.5x model reduction. I have not been able to match these numbers with those in table 2. How do I get to see 7% lower accuracy for BinaryFlex-1.6?

Results suggest a model under 2MB is convenient for using in ARM, is this actually a fact (is it tested in an ARM?) or just guessing? This is also a point made in the introduction and I would expect at least an example of running time there (showing the benefit compared to competitors). It is also interesting the fact that in the text, the ARM is said to have 512KB while in the experiments there is no model achieving that lower-bound.

I would like to see an experiment on ImageNet where the proposed BinaryFlex leads to a model of approximately 7.5MB and see what the preformance is for that model (so comparable in size with the state-of-the-art).

I missed details for the exact implementation for the other datsets (as said in the paper). There are modifications that are obscure and the benefits in model size (at least compared to a baseline) are not mentioned. Why?

---

> ### Author Response · Authors · 2018-01-05
> **Updates**
>
> Thank you for your comments. We have now submitted a new version of our BinaryFlex paper and addressed the issues raised during the reviewing period. We focused on improving clarity and readability as well as providing more results.

---

### Author Response · Authors · 2018-01-05
**Updated paper addressing the issues**

Based on the comment provided by the reviewers, we have made de following modifications to our paper:

- The filter generation stage using OVSF orthogonal binary basis was an important stage in our architecture that wasn't properly explained. We have addressed this problem with a diagram and an explanation in Section 3.1.

- Added result comparing the performance of BinaryFlex on ImageNet, CIFAR-10 and MNIST at different model sizes, i.e. different OVSF ratios.

- Introduced a more readable and informative Section 4. Included a table showing all the parameters used in a give BinaryFlex configuration with 3.4 MB of model size.

- Fixed typos, acronyms usage and figures.

---

### Decision · Program_Chairs · 2018-01-29
**ICLR 2018 Conference Acceptance Decision**

**Decision:**

Reject

**Comment:**

The paper proposes using a set of orthogonal bases that combine to form convolution kernels for CNNs leading to a significant reduction of memory usage. The main concerns raised by the reviewers were 1) clarity; 2) issues with writing and presentation of results; 3) some missing experiments. The authors released a revised version of the paper and a short summary of the enhancements. None of the reviewers changed scores following the author response. The reviews were detailed and came from those familiar with CNNs. I have decided to go with reviewer consensus.